# DeepSTAPLE: Learning to predict multimodal registration quality for unsupervised domain adaptation

Christian Weihsbach, Alexander Bigalke, Christian N. Kruse, Hellena Hempe, and Mattias P. Heinrich

Institute of Medical Informatics, Universität zu Lübeck, Ratzeburger Allee 160, 23538 Lübeck, Germany christian.weihsbach@uni-luebeck.de
https://www.imi.uni-luebeck.de/en/institute.html

**Abstract.** While deep neural networks often achieve outstanding results on semantic segmentation tasks within a dataset domain, performance can drop significantly when predicting domain-shifted input data. Multi-atlas segmentation utilizes multiple available sample annotations which are deformed and propagated to the target domain via multimodal image registration and fused to a consensus label afterwards but subsequent network training with the registered data may not yield optimal results due to registration errors. In this work, we propose to extend a curriculum learning approach with additional regularization and fixed weighting to train a semantic segmentation model along with data parameters representing the atlas confidence. Using these adjustments we can show that registration quality information can be extracted out of a semantic segmentation model and further be used to create label consensi when using a straightforward weighting scheme. Comparing our results to the STAPLE method, we find that our consensi are not only a better approximation of the oracle-label regarding Dice score but also improve subsequent network training results.

**Keywords:** domain adaptation · multi-atlas registration · label noise · consensus · curriculum learning

## 1 Introduction

Deep neural networks dominate the state-of-the-art medical image segmentation [10, 14, 20], but their high performance is depending on the availability of large-scale labelled datasets. Such labelled data is often not available in the target domain and direct transfer learning leads to performance drops due to domain shift [27]. To overcome these issues transferring existing annotations from a labeled source to the target domain is desirable. Mutli-atlas segmentation is a popular method, which accomplishes such a label transfer in two steps: First, multiple sample annotations are transferred to target images via image registration [7, 18, 24] resulting in multiple "optimal" labels [1]. Secondly label fusion can be applied to build the label consensus. Although many methods for

finding a consensus label have been developed [1, 6, 19, 25, 26], the resulting fused labels are still not perfect and exhibit label noise, which complicates the training of neural networks and degrades performance.

**Related work** In the past, various label fusion methods have been proposed, which use weighted voting on registered label candidates to output a common consensus label [1, 6, 19, 26]. More elaborate fusion methods also use image intensities [25], however when predicting across domains source and target intensities can differ substantially complicating intensity-based fusion and would therefore require handling of the intensity gap i.e. with image-to-image translation techniques [29]. When using the resulting consensus labels from non-optimal registration and fusion for subsequent CNN training, noisy data is introduced to the network [12]. Network training can then be improved with techniques of curriculum learning to estimate label noise (i.e. difficulty) and guide the optimization process accordingly [3, 22] but the techniques have not been used in the context of noise introduced through registered pixel-wise labels [2, 3, 11, 22, 28] or employ more specialized and complex pipelines [4, 5, 15]. Other deep learning-based techniques to address ambiguous labels are probabilistic networks [13].

**Contributions** We propose to use data parameters [22] to weight noisy atlas samples as a simple but effective extension of semantic segmentation models. During training the data parameters (scalar values assigned to each instance of a registered label) can estimate the label trustworthiness globally across all multi-atlas candidates of all images. We extend the original formulation of data parameters by additional *risk regularization* and *fixed weighting* terms to adapt to the specific characteristics of the segmentation task and show that our adaptation improves network training performance for 2D and 3D tasks in the single-atlas scenario. Furthermore, we apply our method to the multi-atlas 3D image scenario where the network scores do not improve but yield equal performance in comparison to normal cross-entropy loss training when using out-of-line backpropagation. Nonetheless, we still can achieve an improvement by deriving an optimized consensus label from the extracted weights and applying a straight-forward weighted-sum on the registered atlases.

## 2    Method

In this section, we will describe our data parameter adaption[1] and introduce our proposed extensions when using it in semantic segmentation tasks, namely a special regularization and a fixed weighting scheme. Furthermore, a multi-atlas specific extension will be described, which improves training stability.

---

[1] Our code is openly available on GitHub: https://github.com/multimodallearning/deep_staple

***Data parameters*** Saxena et al. [22] formulate their data parameter and curriculum learning approach as a modification altering the logits input of the loss function. By a learnable logits-weighting improvements could be shown in different scenarios when either noisy training samples and/or classes were weighted during training. Our implementation and experiments focus on per-sample parameters $\mathbf{DP_S}$ of a dataset $S = \{(\mathbf{x_s}, \mathbf{y_s})\}_{s=1}^{n}$ with images $x_s$ and labels $y_s$ containing $n$ training samples. Since weighting schemes for multi-atlas label fusion like STAPLE [26] use a confidence weight of 0 indicating "no confidence" and 1 indicating "maximum confidence" we slightly changed the initial formulation of data parameters:

$$\mathbf{DP}_\sigma = sigmoid\left(\mathbf{DP_S}\right) \tag{1}$$

According to Eq. 1 we limit the data parameters applied to our loss to $DP_\sigma \in (0, 1)$ where a value of 0 indicates "no confidence" and 1 indicates "maximum confidence" such as weighting schemes like STAPLE [26]. The data parameter loss $\ell_{DP}$ is calculated as

$$\ell_{DP}\left(f_\theta\left(\mathbf{x_B}\right), \mathbf{y_B}\right) = \sum_{b=1}^{|B|} \ell_{CE,spatial}\left(f_\theta\left(\mathbf{x_b}\right), \mathbf{y_b}\right) \cdot DP_{\sigma_b} \quad \text{with} \quad B \subseteq S \tag{2}$$

where $B$ is a training batch, $\ell_{CE,spatial}$ is the cross-entropy loss reduced over spatial dimensions and $f_\theta$ the model. As in the original implementation, the parameters require a sparse implementation of the Adam optimizer to avoid diminishing momenta. Note, that the data parameter layer is omitted for inference — inference scores are only affected indirectly by data parameters through optimized model training.

***Risk Regularisation*** Even when a foreground class is present in the image and a registered target label only contains background voxels, the network can achieve a zero-loss value by overfitting. As a consequence, upweighting the overfitted samples will be of no harm in terms of loss reduction which leads to the upweighting of maximal noisy (empty) samples. We therefore add a so called *risk regularisation* encouraging the network to take *risk*

$$\ell = \ell_{DP} - \sum_{b=1}^{|B|} \frac{\#\left\{f_\theta\left(\mathbf{x_b}\right) = c\right\}}{\#\left\{f_\theta\left(\mathbf{x_b}\right) = c\right\} + \#\left\{f_\theta\left(\mathbf{x_b}\right) = \bar{c}\right\}} \cdot DP_{\sigma_b} \tag{3}$$

where $\#\left\{f_\theta\left(\mathbf{x_b}\right) = c\right\}$ and $\#\left\{f_\theta\left(\mathbf{x_b}\right) = \bar{c}\right\}$ indicate positive and negative predicted voxel count. According to this regularisation the network can reduce loss when predicting more target voxels under the restriction that the sample has a high data parameter value i.e. is classified as a clean sample. This formulation is balanced because predicting more positive voxels will increase the cross-entropy term if the prediction is inaccurate.

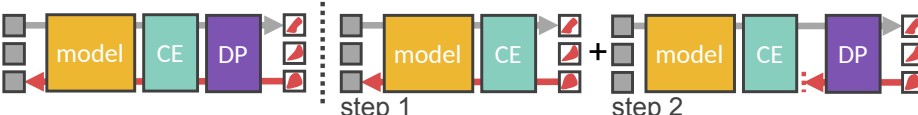

**Fig. 1. Left:** Inline backpropagation updating (red arrow) model and data parameters together. **Right:** Out-of-line backpropagation first steps on model (gray arrow) using normal cross-entropy loss and then steps on data parameters using the model's weights of the first step.

***Fixed weighting scheme*** We found that the parameters have a strong correlation with the ground-truth voxels present in their values. Applying a fixed compensation weighting to the data parameters $DP_{\sigma_b}$ can improve the correlation of the learned parameters and our target scores

$$DP_{\tilde{\sigma}_b} = \frac{DP_{\sigma_b}}{log\left(\#\left\{(\mathbf{y_b} = c\right\} + e\right) + e}$$

(4)

where $\#\left\{\mathbf{y_b} = c\right\}$ denotes the count of ground-truth voxels and $e$ Euler's number.

***Out-of-line backpropagation process for improved stability*** The interdependency of data parameters and model parameters can cause convergence issues when training *inline*, especially during earlier epochs when predictions are inaccurate. We found that a two-step forward-backward pass, first through the main model and in the second step through the main model and the data parameters can maintain stability while still estimating label noise (see Fig. 1). First only the main model parameters will be optimized. Secondly only the data parameters will be optimized *out-of-line*. When using the *out-of-line*, two-step approach data parameter optimization becomes a hypothesis of *"what would help the model optimizing right now?"* without intervening. Due to the optimizer momentum the parameter values still become reasonably separated.

***Consensus generation via weighted voting*** To create a consensus $\mathbf{C_M}$ we use a simple weighted-sum over a set of multi-atlas labels $M$ associated to a fixed image that turned out to be effective

$$\mathbf{C_M} = \left(\sum_{m=1}^{|M|} softmax(\mathbf{DP_M})_m \cdot \mathbf{y_m}\right) > 0.5 \quad \text{with} \quad M \subset S$$

(5)

where $\mathbf{DP_M}$ are the parameters associated to the set of multi-atlas labels $\mathbf{y_M}$.

## 3 Experiments

In this section, we will describe general dataset and model properties as well as our four experiments which increase in complexity up to the successful application of our method in 3D multi-atlas label noise estimation. We will refer to oracle-labels[2] as the real target labels which belong to an image and "registered/training/ground-truth"-labels as image labels that the network used to update its weights. Oracle-Dice refers to the overlapping area of oracle-labels and "registered/training/ground-truth"-labels.

***Dataset*** For our experiments, we chose a challenging multimodal segmentation task which was part of the CrossMoDa challenge [23]. The data contains contrast-enhanced T1-weighted brain tumour MRI scans and high-resolution T2-weighted images (initial resolution of $384/448 \times 348/448 \times 80$ *vox* @ $0.5\ mm \times 0.5\ mm \times 1.0 - 1.5\ mm$ and $512 \times 512 \times 120$ *vox* @ $0.4 \times 0.4 \times 1.0 - 1.5\ mm$). We used the original TCIA dataset [23] to provide omitted labels of the CrossModa challenge which served as oracle-labels. Prior to training isotropic resampling to $0.5\ mm \times 0.5\ mm \times 0.5\ mm$ was performed as well as cropping the data to $128 \times 128 \times 128$ *vox* around the tumour. We omitted the provided cochlea labels and train on binary masks of background/tumour. As the tumour is either contained on the right- or left side of the hemisphere, we flipped the right samples to provide pre-oriented training data and omit the data without tumour structures. For the 2D experiments we sliced the last data dimension.

***Model and training settings*** For 2D segmentation, we employ a LR-ASPP MobileNetV3-Large model [9]. For 3D experiments we use a custom 3D-MobileNet backbone similar as proposed in [21] with an adapted 3D-LR-ASPP head [8]. 2D training was performed with an AdamW [17] optimizer with a learning rate of $\lambda_{2D} = 0.0005$, $|B|_{2D} = 32$, cosine annealing [16] as scheduling method with restart after $t_0 = 500$ batch steps and multiplication factor of 2.0. For the data parameters, we used the SparseAdam-optimizer implementation together with the sparse Embedding structure of PyTorch with a learning rate of $\lambda_{DP} = 0.1$, no scheduling, $\beta_1 = 0.9$ and $\beta_2 = 0.999$. 3D training was conducted with learning rate of $\lambda_{3D} = 0.01$, $|B|_{3D} = 8$ due to memory restrictions and exponentially decayed scheduling with factor of $d = 0.99$. As opposed to Saxena et al. [22] during our experiments we did not find weight-clipping, weight decay or $\ell_2$-regularisation on data parameters to be necessary. Parameters $DP_s$ were initialized with a value of 0.0. For all experiments, we used spatial affine- and b-spline-augmentation and random-noise-augmentation on image intensities. Prior to augmenting we upscaled the input images and labels to $256 \times 256\ px$ in 2D- and $192 \times 192 \times 192$ *vox* in 3D-training. Data was split into 2/3 training and 1/3 validation images during all runs and used global class weights $1/n_{bins}{}^{0.35}$.

---

[2] "The word oracle [...] properly refers to the priest or priestess uttering the prediction.". "Oracle." Wikipedia, Wikimedia Foundation, 03 Feb 2022, en.wikipedia.org/wiki/Oracle

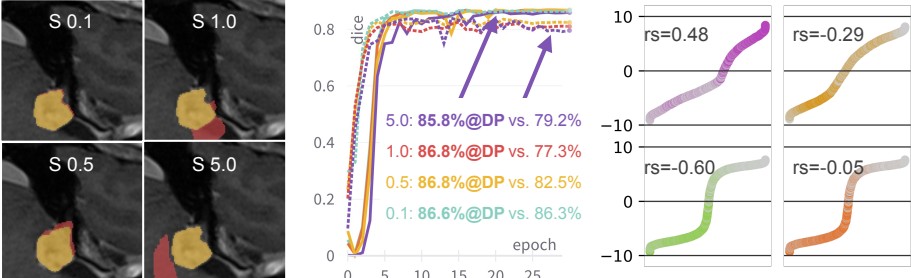

**Fig. 2. Left:** Sample disturbance ■ at strengths [0.1, 0.5, 1.0, 5.0]. **Middle:** Validation Dice when training with named disturbance strenghts, either with data parameters enabled (—) or disabled (- -). **Right:** Parameter distribution for combinations of risk regularization (RR) and fixed weighting (FW): RR+FW ■ | RR ■ | FW ■ | NONE ■. Saturated data points indicate higher oracle-Dice. Value of ranked Spearman-correlation $r_s$ between data parameters and oracle-Dice given.

***Experiment I: 2D model training, artificially disturbed ground-truth labels*** This experiment shows the general applicability of data parameters in the semantic segmentation setting when using one parameter per 2D slice. To simulate label-noise, we shifted 30% of the non-empty oracle-slices with different strengths (Fig. 2, left) to see how the network scores behave (Fig. 2, middle) and whether the data parameter distribution captures the artificially disturbed samples (Fig. 2, right). In case of runs with data parameters the optimization was enabled after 10 epochs.

***Experiment II: 2D model training, quality-mixed registered single-atlas labels*** Extending experiment I, in this setting we train on real registration noise with 2D slices on single-atlases. We use 30 T1-weighted images as fixed targets (non-labelled) and T2-weighted images and labels as moving pairs. For registration we use the deep learning-based algorithm Convex Adam [24]. We select two registration qualities to show quality influence during training: *Best*-quality registration means the single best registration with an average of around 80% oracle-Dice across all atlas registrations. *Combined*-quality means a clipped, gaussian-blurred sum of all 30 registered atlas registrations (some sort of consensus). We then input a mix of 50%/50% randomly selected best/combined labels into training. Afterwards we compare the 100% best, 50%/50% mixed and 100% combined selections focusing on the mixed setting where we train with and without data parameters. Validation scores were as follows (descending): best@no-data-parameters 81.1%, mix@data-parameters 74.1%, mix@no-data-parameters 69.6% and combined@no-data-parameters 61.9%.

***Experiment III: 3D model training, registered multi-atlas labels*** Extending experiment II, in this setting we train on real registration noise but with 3D volumes and multiple atlases per image. We follow the CrossMoDa [23] challenge task and use T2-weighted images as fixed targets (non-labelled) and T1-weighted images and labels as moving pairs. We conducted registration with two algorithms (iterative deeds [7] and deep learning-based algorithm Convex

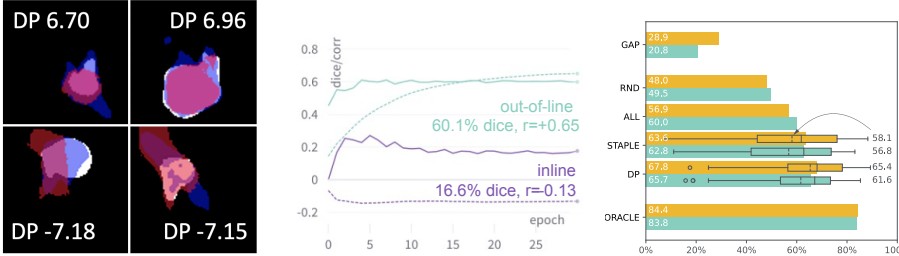

**Fig. 3.** Selected samples with low- and high parameters: Oracle-label □, network prediction ■ and deeds registered label ■

**Fig. 4.** Inline ■ and out-of-line ■ backpropagation. Validation Dice (—) and Spearman-corr. of params. and oracle-Dice (- -)

**Fig. 5.** **FG:** Box plots of STAPLE and DP consensus quality, mean value on the right. **BG:** Bar plot of nnUNet scores; deeds ■, Convex Adam ■

Adam [24]). For each registration method 10 registered atlases per image are fed to the training routine expanding the T2-weighted training size from 40 to 400 label-image pairs each. Fig. 4 shows a run with inline and out-of-line (see Sec. 2) data parameter training on the deeds registrations as an example how training scores behave.

***Experiment IV: Consensus generation and subsequent network training*** Using the training output of experiment III, we built 2x40 consensi: [10 deeds registered @ 40 fixed] and [10 Convex Adam registered @ 40 fixed]. Consensi were built by applying the STAPLE algorithm as baseline and opposed to that our proposed weighted-sum method on data parameters (DP) (see Sec. 2). On these, we trained several powerful nnUnet-models for segmentation [10]. In Fig. 5 in the foreground four box plots show the quality range of generated consensi regarding the oracle dice: [deeds, Convex Adam registrations]@[STAPLE, DP]. In the background the mean validation Dice of nnUnet-model trainings (150 epochs) is shown. As a reference, we trained directly on the T1-moving data with strong data augmentation (nnUNet "insane" augmentation) trying to overcome the domain gap directly (GAP). Furthermore, we trained on 40 randomly selected atlas labels (RND), all 400 atlas labels (ALL), STAPLE consensi, data parameter consensi (DP) and oracle-labels either on deeds or Convex Adam registered data. Note that the deeds data contained 40 unique moving atlases whereas the Convex Adam data contained 20 unique moving atlases, both warped to 40 fixed images as stated before.

## 4    Results and discussion

In **experiment I** we could show that our usage of data parameters is generally effective in the semantic segmentation scenario under artificial label noise. Fig. 2 (middle) shows an increase of validation scores when activating stepping on data parameters after 10 epochs for disturbance strengths > 0.1. Stronger disturbances lead to more severe score drops but can be recovered by using data

parameters. In Fig. 2 (right) one can see that data parameters and oracle-Dice correlate most, when using the proposed risk regularization as well as the fixed weighting-scheme configuration (see Sec. 2). We did not notice any validation score improvements when switching between configurations and therefore conclude that a sorting of samples can also be learned inherently by the network. However, properly weighted data parameters can extract this information, make it explicitly visible and increase explainability. In **experiment II** we show that our approach works for registration noise during 2D training: When comparing different registration qualities, we observed that training scores drop from 81.1% to 69.6% Dice when lowering registration input quality. By using data parameters we can recover to a score of 74.1% meaning an improvement of +4.5%. **Experiment III** covers our target scenario — 3D training with registered multi-atlas labels. With inline training of data parameters (used in the former experiments), validation scores during training drop significantly. Furthermore the data parameters do not separate high- and low quality registered atlases well (see Fig. 4, inline). When using our proposed out-of-line training approach (see Sec. 2) validation Dice and ranked correlation of data parameter values and oracle-Dice improve. **Experiment IV** shows that data parameters can be used to create a weighted-sum consensus as described in Sec. 2: Using data parameters, we can improve mean consensus-Dice for both, deeds and Convex Adam registrations over STAPLE [26] from 58.1% to 64.3% (+6.2%, ours, deeds data) and 56.8% to 61.6% (+4.8%, ours, Convex Adam data). When using the consensi in a subsequent nnUNet training [10], scores behave likewise (see Fig. 5). Regarding training times of over an hour with our LR-ASPP MobileNetV3-Large training, one has to consider that applying the STAPLE algorithm is magnitudes faster.

## 5   Conclusion and outlook

Within this work, we showed that using data parameters in a multimodal prediction setting with propagated source labels is a valid approach to improve network training scores, get insight into training data quality and use the extracted info about sample quality in subsequent steps namely to generate consensus segmentations and provide these to further steps of deep learning pipelines. Our improvements over the original data parameter approach for semantic segmentation show strong results in both 2D- and 3D-training settings. Although we could extract sample quality information in the multi-atlas setting successfully, we could not improve network training scores in this setting directly since using the data parameters inline of the training loop resulted in unstable training. Regarding that, we want to continue investigating how an inline training can directly improve training scores in the multi-atlas setting. Furthermore our empirically chosen fixed weighting needs more theoretical foundation. The consensus generation could be further improved by trying more complex weighting schemes or incorporating the network predictions itself. Also we would like to compare our registration-segmentation pipeline against specialized approaches of Ding et al. and Liu et al. [4, 5, 15] which we consider as very interesting baselines.

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
