# OpenReview forum: "DeepSTAPLE: Learning to predict multimodal registration quality for unsupervised domain adaptation "
_WBIR.info/2022/Workshop/Biomedical_Imaging_Registration — WBIR 2022_

### Official Review · Reviewer_YRun · 2022-02-17

**Rating:** 2
**Confidence:** 1

**Deanonymize Review:**

no

**Detailed Comments:**

I would advise the authors to improve the readability of the paper to clearly and precisely explain the presented concepts and contributions. I cannot validate the validity and novelty based on the presentation in its present state.

**Paper Type:**

both

**Strengths Weaknesses:**

The paper concerns neural network based image segmentation. The authors proposed to introduce regularization and weighting in curriculum learning to account for uncertainty in labels coming from the atlas registration process. The authors test their method on both 2D and 3D datasets.

Strengths:
* accounting for label uncertainty can potentially improve the training of neural networks and thereby the segmentation results
* the authors explain the motivation for their work and the contributions clearly in the introduction

Weaknesses:
* I find the description of the proposed method very hard to read and imprecisely described. In section 2, the background for the method is not properly introduced and many terms in the mathematical notation are not defined. Based on the description, I cannot evaluate to which degree the method is sound and novel
* while the authors perform extensive experiments, it is very hard to properly validate the experiments and results from the text in section 3 and 4

---

### Official Review · Reviewer_HsLN · 2022-02-21

**Rating:** 4
**Confidence:** 4
**Recommendation:** Short Oral

**Deanonymize Review:**

no

**Detailed Comments:**

I would like the authors to explain what happens to the data parameters during the inference phase.

**Paper Type:**

both

**Strengths Weaknesses:**

The paper proposes a learning approach applied to multi-modal semantic segmentation problems. During training, the segmentation model incorporates a set of learnable parameters to handle noisy samples and improve subsequent estimations. The authors evaluate their method in different 2D and 3D cross-domain atlas-based segmentation scenarios.

The paper is well written and the ideas are clearly expressed. The experiments are diverse and the results show that the approach helps to improve the performance of segmentation models. I think it is a very interesting idea and has a lot of room for improvement in the future.

---

### Official Review · Reviewer_aAM1 · 2022-02-22

**Rating:** 4
**Confidence:** 4
**Recommendation:** Long Oral, Short Oral

**Deanonymize Review:**

no

**Detailed Comments:**

In general, the paper is well-written and well-structured. The idea of learning the quality of multimodal registration using data parameters in the context of multi-atlas segmentation is interesting. The experiments and results nicely show the potential of the method.

Some minor comments:
- Saxena et. al.: not point after et.
Please also provide the reference number [22]
- “By correlating the data parameter values with our target scores in preliminary experiments, we found that the parameters have a strong correlation with the ground-truth pixels present in their values.”
I don’t know what that means. Ground truth pixels in their values?

**Paper Type:**

methodological development

**Strengths Weaknesses:**

Strength
- This is a well-written and well-structured paper that proposes to learn multimodal registration quality using ‘data parameters’. Such learned quality is then used in a weighted voting scheme for multi-atlas label fusion.
- Extensions to the data parameter framework of Saxena et al. are proposed: risk regularization, fixed weighting scheme and out-of-line backpropagation.
- The experiments in 2D and 3D are appropriate to show the effectiveness of the method.

Weaknesses
- The idea of the work is heavily based on [Saxena et al., 2019] (reference [22] of the manuscript), where the data parameters are introduced. This has to be made very clear in the introduction. The reference is for example missing in the paragraph about the contribution.
- There are inaccuracies in the notation, not all variables are sufficiently explained, which makes it difficult to follow. For example: what is f_theta, z? The network output and reference? The circle with a dot is the Hadamard product?
- The description of the experiments is difficult to follow. In the beginning it is unclear what the purpose of the different experiments is, which only comes clearer when describing the results. I would advise to clearly state the overall objective of each experiment from the beginning of Sec. 3 on.
- There are some repetitions in the introduction, mainly in first and second paragraph.

---

### Official Review · Reviewer_kXSM · 2022-02-22

**Rating:** 3
**Confidence:** 4

**Deanonymize Review:**

no

**Detailed Comments:**

• I think the discussion of the results would be easier to follow in case the authors summarize them into a table
• Some more metrics except dice coefficient should be added to assess the performance of the different configurations.

**Paper Type:**

validation / application paper

**Strengths Weaknesses:**

Strengths

 • The paper presents the use of medical image registration as a way to effectively extend the semantic segmentation models.

 • Experiments had been constructed in both 2D and 3D tasks in the single-atlas scenario as well as multi-atlas 3D.


Weaknesses

• The paper does not provide some methodological novelties in terms of registration methods. However, it demonstrates an application to domain adaptation.

 • The paper misses some comparisons with other methods for the unsupervised domain adaptation task. In the current form, it is difficult to evaluate the contributions of the paper.

 • The results section and the discussion of the different experiments are difficult to follow.

 • The authors used deeds framework as a registration method. I would be interested to see what would be the performance of the method using different registration frameworks or even some deep learning-based ones.

---

### Decision · Program_Chairs · 2022-02-22

Accept